# Relationship between Serum 25-Hydroxyvitamin D Level and Risk of Recurrent Stroke

**DOI:** 10.3390/nu14091908

**Published:** 2022-05-02

**Authors:** Guowei Li, Likang Li, Jonathan D. Adachi, Ruoting Wang, Zebing Ye, Xintong Liu, Lehana Thabane, Gregory Y. H. Lip

**Affiliations:** 1Center for Clinical Epidemiology and Methodology (CCEM), Guangdong Second Provincial General Hospital, Guangzhou 510317, China; lilikangccem@hotmail.com (L.L.); wangruoting1996@163.com (R.W.); 2Department of Health Research Methods, Evidence, and Impact (HEI), McMaster University, Hamilton, ON L8S 4L8, Canada; thabanl@mcmaster.ca; 3Department of Medicine, McMaster University, Hamilton, ON L8S 4L8, Canada; jd.adachi@sympatico.ca; 4Department of Cardiology, Guangdong Second Provincial General Hospital, Guangzhou 510317, China; tgccem@hotmail.com; 5Department of Neurology, Guangdong Second Provincial General Hospital, Guangzhou 510317, China; jzhzang@163.com; 6Centre for Evaluation of Medicines, St. Joseph’s Healthcare, Hamilton, ON L8N 4A6, Canada; 7Liverpool Centre for Cardiovascular Science, University of Liverpool, Liverpool L69 3BX, UK; gregory.lip@liverpool.ac.uk; 8Aalborg Thrombosis Research Unit, Department of Clinical Medicine, Aalborg University, 9000 Aalborg, Denmark

**Keywords:** vitamin D, recurrent stroke, 25-hydroxyvitamin, stroke prevention

## Abstract

Evidence for the association between vitamin D and risk of recurrent stroke remains sparse and limited. We aimed to assess the relationship between serum circulating 25-hydroxyvitamin D (25(OH)D) level and risk of recurrent stroke in patients with a stroke history, and to identify the optimal 25(OH)D level in relation to lowest recurrent stroke risk. Data from the nationwide prospective United Kingdom Biobank were used for analyses. Primary outcome was time to first stroke recurrence requiring a hospital visit during follow-up. We used Cox proportional hazards regression model with restricted cubic splines to explore 25(OH)D level in relation to recurrent stroke. The dose-response relationship between 25(OH)D and recurrent stroke risk was also estimated, taking the level of 10 nmol/L as reference. A total of 6824 participants (mean age: 60.6 years, 40.8% females) with a baseline stroke were included for analyses. There were 388 (5.7%) recurrent stroke events documented during a mean follow-up of 7.6 years. Using Cox proportional hazards regression model with restricted cubic splines, a quasi J-shaped relationship between 25(OH)D and risk of recurrent stroke was found, where the lowest recurrent stroke risk lay at the 25(OH)D level of approximate 60 nmol/L. When compared with 10 nmol/L, a 25(OH)D level of 60 nmol/L was related with a 48% reduction in the recurrent stroke risk (hazard ratio = 0.52, 95% confidence interval: 0.33–0.83). Based on data from a large-scale prospective cohort, we found a quasi J-shaped relationship between 25(OH)D and risk of recurrent stroke in patients with a stroke history. Given a lack of exploring the cause–effect relationship in this observational study, more high-quality evidence is needed to further clarify the vitamin D status in relation to recurrent stroke risk.

## 1. Introduction

Stroke is a serious public health concern, especially with the aging population, ranking as the second leading cause of death worldwide [1]. A previous stroke is substantially related with increased risk of subsequent stroke, with a recurrent rate ranging from 5% to 40% within five years [2,3,4]. Hence, there has been a move towards a more holistic and integrated care approach to stroke management, including attention to lifestyle and comorbidities [5].

Recently, vitamin D has been widely found to associate with risk of stroke, based on data from observational studies. Several meta-analyses consistently reported that low level of circulating 25-hydroxyvitamin D (25(OH)D) was significantly related with the onset of stroke [6,7,8,9]. Nevertheless, evidence for the association between vitamin D and risk of *recurrent* stroke remains sparse and limited. Some studies reported that among patients with a previous stroke, those with the first quartile of 25(OD)D level had a significantly highest risk of recurrent stroke [10,11]. However, their small sample sizes and short-term follow-up precluded further investigation of adequate vitamin D levels in relation to stroke recurrence. Furthermore, the optimal level of 25(OH)D associated with the lowest risk of recurrent stroke remains largely unknown, especially given the non-linear relationship between 25(OH)D level and stroke risk.

Therefore, in this study, we aimed to assess the association between serum 25(OH)D level and risk of recurrent stroke in patients with a prior stroke history, and second, to identify the optimal 25(OH)D level in relation to lowest recurrent stroke risk. Data from the nationwide prospective United Kingdom (UK) Biobank were used for analyses.

## 2. Methods

### 2.1. Participants and Setting

Descriptions about the UK Biobank have been published in the literature and on the website (www.ukbiobank.ac.uk, accessed on 10 December 2021) [12]. In brief, the UK Biobank is a nationwide prospective cohort study that recruited >500,000 community dwellers between year 2006 and 2010, with the goals of improving diagnosis, prevention, treatment, and prognosis of diseases for the middle-aged and older adults. The study collected data from participant self-reports, interview with trained staff and physical measures, and used multiple data sources for linkage. The UK Biobank was approved by the Northwest Multicenter Research Ethics committee (11/NW/0382). The Guangdong Second General Provincial Hospital Research Ethics Committee approved the current analysis (2022-KY-KZ-119-01). All participants provided written informed consent.

For our current study, we limited eligible participants to those with a history of stroke at baseline. Information on history of stroke was identified from patient self-reports, the ICD-10 and ICD-9 code at baseline. The patient selection process is displayed in Appendix A for this study.

### 2.2. Outcome Measures

Our primary outcome was time to first stroke recurrence requiring a hospital visit during follow-up. The ICD-10 codes were used to determine the recurrent stroke events and their corresponding survival time (Appendix A shows the codes used for stroke identification). Secondary outcomes included ischemic and hemorrhagic stroke. Patients were followed up to stroke recurrence, 31 March 2017, or death, whichever came first.

### 2.3. Serum 25(OH)D Levels and Other Independent Variables

Serum 25(OH)D level (in nmol/L) was measured from the non-fasted blood sample drawn at the time of study enrollment, using the Liaison XL 25(OH)D assay.

Data on other independent variables at baseline included age, sex, smoking and drinking, ethnicity, education, body mass index (BMI), physical activity, atrial fibrillation, hypertension, heart failure, hypercholesterolemia, and diabetes. We also collected information on intake of statins, non-steroidal anti-inflammatory drugs (NSAIDs), anticoagulants, antihypertensive and antidiabetic medications, and mineral and vitamin supplementation. To enhance the under-recognition of data on comorbidities and medication intake at baseline, we used the information from patient’s self-reports, interviews with trained staff regarding medications and treatment that patients received, and ICD codes. We documented the existence of a variable if the patient had a positive response to any of the aforementioned data fields.

## 3. Statistical Analysis

Continuous and categorical variables were depicted with mean (standard deviation, SD) and frequency (percentage), respectively. Comparisons of baseline information between patients with and without recurrent stroke were performed by t-test for continuous variables and chi-square test for categorical variables. We used the Kaplan–Meier method to graph failure curve for recurrent stroke. Kernel density estimation was used to estimate the probability density of participants’ serum 25(OH)D level.

To illustrate the relationship between 25(OH)D level and risk of recurrent stroke, we conducted a Cox proportional hazards regression model that was adjusted for the baseline characteristics as listed in Table 1 including age, sex, BMI, smoking and drinking, physical activity, comorbidities, medications, and supplementation. Results were presented with hazard ratios (HRs) and corresponding 95% confidence intervals (CIs). To explore the optimal of 25(OH)D level, we employed the restricted cubic splines with knots at 5th, 35th, 65th, and 95th percentiles to smooth the non-linear association between 25(OH)D level and recurrent stroke risk. We also showed the dose-response relationship between 25(OH)D level and recurrent stroke risk to estimate the HRs for pre-determined levels at 20, 30, 40, 50, 60, 70, and 80 nmol/L, taking the 25(OH)D of 10 nmol/L as reference. Similar analyses were also conducted for secondary outcomes of ischemic and hemorrhagic stroke.

We performed two *post hoc* sensitivity analyses to assess the robustness of the main findings. A competing risk analysis using the Fine and Gray model was conducted for the relationship between 25(OH)D levels and risk of recurrent stroke, by treating all-cause death as competing events for recurrent stroke. Recognizing that the 25(OH)D levels may have oscillation by months and the length between the onset of stroke and blood sampling date may be different, we performed another sensitivity analysis in the multivariable Cox model after further adjusted for these two variables (the month for 25(OH)D measures, and the length between the stroke onset and blood sampling date).

Two pre-defined subgroups were conducted by sex (males and females) and age (<65 and ≥65 years) to explore whether there existed potential effect modifications. Furthermore, to directly compare with results from previous studies, we estimated the associations between different quartiles of 25(OH)D level and recurrent stroke risk, with the first quartile as reference value. Likewise, we assessed different 25(OH)D levels in relation to recurrent stroke risk using the recognized cut-off points for vitamin D sufficiency (>50 nmol/L), insufficiency (25–50), and deficiency (<25), taking vitamin D deficiency as reference.

All tests were two-sided, and the significance level was set as 0.05. Stata version 17 (StataCorp, College Station, TX, USA) and R version 3.5.1 (R Foundation for Statistical Computing, Vienna, Austria) were used for analyses.

## 4. Results

There were 6824 participants (mean age: 60.6 years, 40.8% females) with a baseline stroke included in this analysis. There were approximately 40% and 7% of participants who never smoked and consumed alcohol, respectively. The majority of participants was physically active (73%) and with hypertension (62%). Other information on comorbidities, medication, and supplementation intake was also presented in Table 1. The mean 25(OH)D level was 46.5 nmol/L.

During a mean follow-up of 7.6 (SD = 1.8) years, there were 388 (5.7%) recurrent stroke events documented including 250 ischemic, 87 hemorrhagic, and 51 unspecified stroke (Appendix A displays the Kaplan–Meier failure curve for recurrent stroke). Table 1 displays comparisons of baseline characteristics between participants with and without recurrent stroke. Participants with recurrent stroke were older, and less likely to be females and physically active when compared with controls (all *p*-values < 0.05). Significantly higher percentages of hypertension and diabetes were found in patients with recurrent stroke. Participants in the recurrent stroke group were more likely to consume statins, anticoagulants, antihypertensive, and antidiabetic medications. Appendix A shows the probability density function of serum 25(OH)D level stratified by participants with and without recurrent stroke. A lower 25(OH)D level was found in patients with recurrent stroke (45.6 vs. 46.5); however, the difference was not statistically significant (*p*-value = 0.48; Table 1).

A quasi J-shaped relationship between 25(OH)D and risk of recurrent stroke was found in this study (Figure 1), where the lowest recurrent stroke risk lay at the 25(OH)D level of 58.2 nmol/L. As Table 2 presents, when taking the 25(OH)D of 10 nmol/L as reference, a 25(OH)D level of 60 nmol/L was related with a 48% reduction in the recurrent stroke risk (HR = 0.52, 95% CI: 0.33–0.83). A total of 658 (9.6%) deaths occurred before recurrent stroke; therefore, these deaths were the competing events for stroke recurrence. The competing risk analysis by treating deaths as competing events yielded similar findings to the main results, with the potential lowest recurrent stroke risk at the 25(OH)D level of approximate 60 nmol/L (subhazards ratio = 0.66, 95% CI: 0.40–1.12; Appendix A). Another sensitivity analysis by further adjusted for the month for 25(OH)D measures, the length between the onset of stroke and blood sampling date, and these two variables, showed similar results to the main findings, with the smallest HR of 0.51, 0.51, and 0.50 correspondingly at the 25(OH)D level of 60 nmol/L (Appendix A).

Analyses for secondary outcomes and by subgroup yielded in general similar results to the main findings, with the lowest smallest recurrent stroke risk at 25(OH)D of about 60 nmol/L (HRs ranging from 0.43 to 0.70); however, the potentially lowest risks of hemorrhagic stroke (HR = 0.38) and for females (HR = 0.29) were observed at approximately 40 nmol/L (Table 2; Appendix A).

Table 3 shows results from additional analyses for the relationship between 25 (OH)D and recurrent stroke risk. When compared with the first quartile, the third quartile of 25(OH)D level (ranging from 43.5 to 61.3 nmol/L) was found to significantly associate with a 32% reduction in recurrent stroke risk (HR = 0.68, 95% CI: 0.48–0.96). Participants with insufficient (HR = 0.60) or sufficient 25(OH)D levels (HR = 0.59) had a significantly and similarly reduced risk of recurrent stroke, when taking 25(OH)D deficiency as reference.

## 5. Discussion

In this nationwide prospective cohort study, we explored what was the optimal vitamin D level in relation to risk of recurrent stroke in patients with a prior stroke history. There was a quasi J-shaped relationship between 25(OH)D and risk of recurrent stroke observed, with the lowest risk found at 25(OH)D level of approximate 60 nmol/L. When compared with 10 nmol/L, a 25(OH)D level of 60 nmol/L was significantly associated with a 48% reduction in recurrent stroke risk.

The majority of literature had consistently found that low 25(OH)D levels were linearly or non-linearly associated with risk of *first-ever* stroke, although the relationship remained controversial. In this study, we found the optimal 25(OH)D level lay at approximate 60 nmol/L regarding the risk of *recurrent* stroke, which was in line with the quasi J-shaped association found from a recent meta-analysis that reported the 25(OH)D level of 50 nmol/L was related with the lowest stroke risk [8]. Even though with a different inflection, our study again confirmed that either a low or high level of 25(OH)D was related with elevated risk of recurrent stroke. Our different optimal 25(OH)D value from the published meta-analysis may be due to different population (with stroke history versus free from stroke) and outcome (stroke recurrence versus onset of stroke), data sources (individual patient data versus published summary data) and study settings (single country versus multi-country). Nevertheless, our results may provide some evidence about the vitamin D status in relation to risk of recurrent stroke.

The role of 25(OH)D level in stroke recurrence was largely remained uninvestigated. Vitamin D status may be associated with stroke size and disease severity, which could subsequently impact the propensity towards stroke recurrence [13,14]. Unfortunately, there were no data on NIHSS (National Institutes of Health Stroke Scale) to evaluate the stroke severity and size between patients with and without recurrent stroke. However, our additional analyses showed similar results to previous studies that further adjusted for NIHSS (Table 3) [10,11]. Vitamin D deficiency had been linked with cardiovascular risk including hypertension, diabetes mellitus, and arterial stiffness, thereby contributing to enhanced stroke risk [15,16,17]. Moreover, vitamin D may own neuroprotective effects, while vitamin D deficiency could promote inflammation and vascular remodelling to increase the risk of stroke [18]. Indeed, in this study, when compared with low level of 25(OH)D or vitamin D deficiency, a decreased risk of recurrent stroke was consistently found. Moreover, high dosages of vitamin D administration in animal experiments could result in widespread arterial calcification, especially when with co-existing diabetes, atherosclerosis, and kidney disease [19]. This also in part supported the elevated risk of recurrent stroke with high 25(OH)D levels (Figure 1). Nevertheless, given the non-randomization design, whether the observed 25(OH)D level was a surrogate for healthy lifestyle and/or frailty status, and whether there was residual biases and confounding effect existed in this study, remained uncertain. Therefore, our findings regarding the relationship between 25(OH)D level and recurrent stroke risk should be interpreted with caution, requiring further high-quality evidence and ideally from randomized controlled trials for clarification and verification.

There have been four studies evaluating the association between 25(OH)D level and risk of recurrent stroke in the literature, three from China [10,11,20] and one from the US [21]. All these studies categorized 25(OH)D level as either four quartiles [10,11] or binary for analyses [20,21]. These studies failed to comprehensively assess vitamin D status in relation to recurrent stroke. Importantly, they did not consider the potential association between high 25(OH)D levels and increased recurrent stroke risk, which would mislead the audience about the incremental beneficial effect of high vitamin D status. Their various cut-off points for 25(OH)D categorization used for analyses could also make their findings difficult to interpret, as it may not be appropriate to use quartiles or dichotomization for primary analyses regarding the J-shaped relationship given the substantial heterogeneity in the defined subgroups. For instance, Huang et al. used the fourth quartile (>22.8 ng/mL, i.e., >57 nmol/L) as a reference group [11], while our results demonstrated a slight decrease followed by continuous increase in recurrent stroke risk with elevated 25(OH)D levels in the group of >57 nmol/L (Figure 1). Moreover, their small sample sizes (ranging from 220 to 946) prohibited further attempts at exploring stroke subtypes and subgroup effects.

By contrast, our study used data from a large-scale prospective cohort with a follow-up of approximate eight years for analyses. The relationship between 25(OH)D and recurrent stroke risk was depicted by graphs from restricted cubic splines in combination with estimates of multiple individual 25(OH)D points, generating a detailed non-linear association between 25(OH)D in relation to recurrent stroke. Furthermore, we performed analyses for stroke subtypes and stratified by age and sex to test the potential effect modifications. Of note, the relationship for hemorrhagic stroke and females seemed to demonstrate a different pattern from the main analysis (Appendix A). Part of the interpretation may be due to the relatively small sample size of recurrent stroke events in females and for hemorrhagic stroke (Table 2). Nonetheless, these exploratory results for different subgroups required further adequately powered and well-designed studies for further investigation and clarification.

## 6. Strength and Limitations

This study used data from a nationwide cohort to comprehensively assess serum 25(OH)D level in relation to recurrent stroke, with results shown as illustrations and estimates from specific 25(OH)D values. Rigorous methodology and robust analyses also supported the study findings. The high risk of stroke recurrence and its substantial impact on mortality remained a severe public health concern [22,23]. While there was an evidence gap in the existing guidelines regarding 25(OH)D level in relation to recurrent stroke, our findings might highlight the importance of adequate vitamin D status in relation to recurrent stroke risk, although we had limited data to explore the causal mechanisms.

Several limitations exited in this study. First, we could not fully preclude confounding effects especially of those unmeasured variables in this observational study, which may compromise the validity and strength of our results [24]. For instance, due to lack of data on NIHSS, whether and to what extent the relationship between vitamin D status and recurrent stroke risk could be influenced by stroke severity and size, remained unknown. Likewise, the relationship between 25(OH)D and recurrent stroke risk may be driven by some unmeasured factors associated with lifestyle and frailty, which would impair our observed findings. Some baseline comorbidities (hypertension, diabetes, and heart failure) and medications (NSAIDs, antihypertensive, and antidiabetic drugs) were significantly associated with serum 25(OH)D levels. Even though we adjusted for all the comorbidities and medications in the multivariable model, unquantified moderator and confounding effects would remain. It was reported that the use of Liaison assay would systematically underestimate the values of 25(OH)D, which may lead to study populations being misclassified as vitamin D deficiency [25]. Therefore, our results should be interpreted with caution, especially regarding the absolute 25(OH)D values and the inflection points found from the quasi J-shaped relationship curves. We only had data on baseline 25(OH)D measures; thus, no analysis for the change in vitamin D status in relation to recurrent stroke could be performed. It would be a worthwhile endeavor to further explore the change in vitamin D status and its potential usefulness for stroke prognosis and risk evaluation. Furthermore, it was uncertain about whether the splines for hemorrhagic stroke and in females were because of either insufficient statistical power or the true absence of a shaped relationship with 25(OH)D. Collectively, our findings from an observational study were primarily hypothesis generating with an exploratory nature, which warranted further exploration to clarify the vitamin D status in relation to risk of recurrent stroke.

## 7. Conclusions

Based on data from a large-scale prospective cohort, we found a quasi J-shaped relationship between 25(OH)D and risk of recurrent stroke in patients with a stroke history, which might provide some insights into the vitamin D status for recurrent stroke prevention. Given a lack of exploring the cause–effect relationship in this observational study, more high-quality evidence is needed to further clarify the vitamin D status in relation to recurrent stroke risk.

## Figures and Tables

**Figure 1 nutrients-14-01908-f001:**
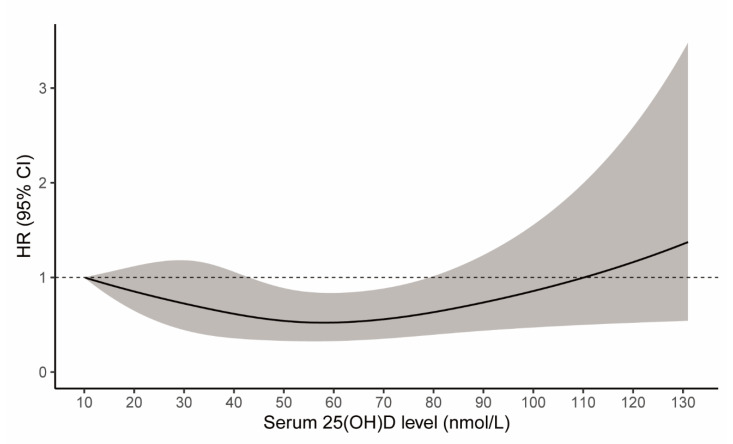
Restricted cubic spline showing the 25(OH)D levels in relation to risk of recurrent stroke (shadow indicating the 95% confidence intervals).

**Table 1 nutrients-14-01908-t001:** Patients’ baseline characteristics and comparisons between patients with and without recurrent stroke.

Characteristics	Total Participants (n = 6824)	Stroke Recurrence during Follow-Up		
		Yes (n = 388)	No (n = 6436)	*p*-Value
**Age:** mean (SD), years	60.6 (6.9)	61.9 (6.4)	60.6 (6.9)	<0.01
**Female sex:** n (%)	2783 (40.8)	136 (35.1)	2647 (41.1)	0.02
**BMI:** mean (SD), kg/m^2^	28.8 (5.1)	28.5 (4.9)	28.9 (5.1)	0.13
**Smoking:** n (%)				
Never	2747 (40.6)	140 (36.4)	2607 (40.9)	0.05
Former	2915 (43.1)	167 (43.4)	2748 (43.1)	
Current	1097 (16.2)	78 (20.3)	1019 (16.0)	
**Alcohol drinking:** n (%)				
Never	446 (6.6)	28 (7.3)	418 (6.5)	0.83
Former	588 (8.7)	32 (8.3)	556 (8.7)	
Current	5761 (84.8)	326 (84.5)	5435 (84.8)	
**White ethnicity:** n (%)	6465 (95.3)	367 (94.8)	6098 (95.3)	0.68
**With college or university degree:** n (%)	567 (8.5)	31 (8.2)	536 (8.5)	0.83
**Physical activity** (≥600 MET min per week): n (%)	3792 (73.1)	196 (66.9)	3596 (73.5)	0.01
**Comorbidity:** n (%)				
Atrial fibrillation	535 (7.8)	39 (10.1)	496 (7.7)	0.10
Hypertension	4222 (61.9)	269 (69.3)	3953 (61.4)	<0.01
Hypercholesterolemia	2996 (43.9)	188 (48.5)	2808 (43.6)	0.06
Diabetes	1003 (14.7)	92 (23.7)	911 (14.2)	<0.01
Heart failure	206 (3.0)	17 (4.4)	189 (2.9)	0.11
**Medication and supplementation intake:** n (%)				
NSAIDs	694 (10.2)	38 (9.8)	656 (10.2)	0.80
Antihypertensive drugs	4075 (59.8)	259 (66.8)	3816 (59.4)	<0.01
Antidiabetic drugs	763 (11.2)	76 (19.6)	687 (10.7)	<0.01
Statins	4462 (65.4)	274 (70.6)	4188 (65.1)	0.03
Anticoagulants	631 (9.2)	51 (13.1)	580 (9.0)	<0.01
Vitamins	1934 (28.7)	110 (28.6)	1824 (28.7)	0.99
Minerals and other dietary supplementation	2685 (39.6)	144 (37.5)	2541 (39.7)	0.39
**Serum 25(OH)D:** mean (SD), nmol/L	46.5 (22.4)	45.6 (25.9)	46.5 (22.1)	0.48

SD = standard deviation; BMI = body mass index; NSAIDs = non-steroidal anti-inflammatory drugs.

**Table 2 nutrients-14-01908-t002:** Results for the relationship between serum 25(OH)D level and recurrent stroke risk.

Outcome/Analysis		No. of Events/No. of Patients	25(OH)D Level, in nmol/L ^1^							
			10	20	30	40	50	60	70	80
Main analysis										
Primary outcome	**Total recurrent stroke**	388/6824	Ref	0.85 (0.65–1.12)	0.72 (0.44–1.18)	0.62 (0.36–1.06)	0.54 (0.33–0.89)	0.52 (0.33–0.84)	0.56 (0.35–0.88)	0.63 (0.39–1.01)
Secondary outcome	**Ischemic stroke**	250/6824	Ref	0.91 (0.64–1.28)	0.82 (0.44–1.54)	0.75 (0.37–1.51)	0.70 (0.37–1.32)	0.69 (0.38–1.26)	0.72 (0.40–1.31)	0.79 (0.43–1.44)
	**Hemorrhagic stroke**	87/6824	Ref	0.64 (0.36–1.13)	0.44 (0.16–1.21)	0.38 (0.12–1.16)	0.40 (0.15–1.09)	0.42 (0.16–1.06)	0.40 (0.16–1.02)	0.38 (0.14–1.02)
Subgroup analysis										
By sex	**Males**	252/4041	Ref	1.01 (0.73–1.41)	0.99 (0.54–1.80)	0.88 (0.45–1.72)	0.74 (0.40–1.37)	0.70 (0.39–1.25)	0.76 (0.43–1.35)	0.91 (0.51–1.63)
	**Females**	136/2783	Ref	0.54 (0.33–0.88)	0.33 (0.14–0.78)	0.29 (0.11–0.72)	0.32 (0.14–0.74)	0.34 (0.16–0.74)	0.31 (0.14–0.67)	0.25 (0.11–0.60)
By age	**<65 years**	205/4355	Ref	0.85 (0.58–1.24)	0.71 (0.36–1.37)	0.56 (0.28–1.12)	0.45 (0.24–0.85)	0.43 (0.23–0.79)	0.47 (0.25–0.86)	0.56 (0.30–1.04)
	**≥65 years**	183/2469	Ref	0.86 (0.58–1.28)	0.75 (0.36–1.57)	0.68 (0.28–1.64)	0.66 (0.29–1.48)	0.66 (0.31–1.39)	0.68 (0.33–1.41)	0.72 (0.34–1.52)

Ref = reference; ^1^ Results shown as hazard ratios (95% confidence intervals) from the models that used restricted cubic splines and were adjusted for age, sex, BMI, smoking and drinking, physical activity, comorbidities, medications, and supplementation.

**Table 3 nutrients-14-01908-t003:** Result from additional analyses for the relationship between 25 (OH)D and recurrent stroke risk.

Serum 25(OH)D Level	Recurrent Stroke		
	No. of Events/No. of Patients	HR (95% CI) ^1^	*p*-Value
**Defined by quartile ^2^**			
1st quartile	117/1719	Ref	-
2nd quartile	92/1707	0.77 (0.56–1.07)	0.12
3rd quartile	86/1693	0.68 (0.48–0.96)	0.03
4th quartile	93/1705	0.77 (0.55–1.08)	0.13
**Defined by status**			
Deficiency (<25 nmol/L)	97/1239	Ref	-
Insufficiency (25–50 nmol/L)	149/2906	0.60 (0.44–0.81)	< 0.01
Sufficiency (>50 nmol/L)	142/2679	0.59 (0.43–0.82)	< 0.01

HR = hazard ratio; CI = confidence interval; Ref = reference; ^1^ Results from the models that used restricted cubic splines and were adjusted for age, sex, BMI, smoking and drinking, physical activity, comorbidities, medications, and supplementation. ^2^ The cut-off points to define quartiles were 28.9, 43.5, and 61.3 nmol/L.

## Data Availability

The data can be available on application to the UK Biobank (www.ukbiobank.ac.uk/). Data described for the analyses and in the manuscript will be made available upon request.

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
