# Peer review of "Relationship between Serum 25-Hydroxyvitamin D Level and Risk of Recurrent Stroke"

_nutrients, 2022, doi:10.3390/nu14091908_

Round 1

Reviewer 1 Report

The manuscript: " Relationship between serum 25-hydroxyvitamin D level and risk of recurrent stroke" is a very interesting and important study to investigate if recurrent stroke is howesome related to levels of serum 25 (OH) D. the analysis in clear presented and well done. I have only a  minor question: there are any relationship ore association between clinical variables (such as modbidities and drugs) and serum (OH)D levels?

Author Response

We thank the reviewer for his/her positive summary and helpful comments.

Based on the comment, after assessing the relationship between baseline 25(OH)D and comorbidities and medications, we found that some comorbidities (hypertension, diabetes, and heart failure) and medications (NSAIDs, antihypertensive and antidiabetic drugs) were significantly related to 25(OH)D levels (all p-values < 0.05 from the Spearman’s correlation test).

Even though we had tried to adjust for the comorbidities and medications in the multivariable model, there may be unquantified moderator and confounding effects on the relationship between 25(OH)D and recurrent stroke risk. We emphasized it as a key limitation in Discussion (Page 10, Paragraph 2; changes highlighted in Blue).

Reviewer 2 Report

Overall: The authors found a J-shaped relationship between 25(OH)D and risk of recurrent stroke in patients with a stroke history, where the lowest recurrent stroke risk lay at the 25(OH)D level of approximate 60 nmol/L. This study used enough number of patients’ data from UK Biobank. This study is very interesting.

Major points:

  1. The primary outcome was time to first stroke recurrence requiring a hospital visit during follow-up. All-cause death can be competing risk. This should be considered in statistical analysis.
  2. Serum levels of 25OHD may have oscillation by months, e.g., high in August and low in February. This should be adjusted.
  3. Patients with a stroke history were sentried in this study. However, there are no data of days between the stroke date and blood sampling date in order to measure serum levels of 25(OH)D. Can these days be a confounder or bias in the analyses of recurrence risks.

Major points:

  1. The primary outcome was time to first stroke recurrence requiring a hospital visit during follow-up. However, all-cause death can be competing risk. This should be considered in statistical analysis and needed to revise.
  2. For survival analyses, is day zero blood sampling? 

Author Response

We appreciate the reviewer’s thoughtful comments on our manuscript.

1.We have performed a post hoc sensitivity analysis by treating all-cause death as competing events for recurrent stroke. Similar findings were found to the main results (Page 5, Paragraph 3; Page 6, Paragraph 4; Page 7, Paragraph 1; Supplemental Table 2; changes highlighted in Blue).

2/3. We agree with the reviewer that the two variables (the month for 25(OH)D measures, and the length between the stroke onset and blood sampling date) should be adjusted in the model. We conducted a sensitivity analysis by further adjusting for the month for 25(OH)D measures, the length between the onset of stroke and blood sampling date, and these two variables (Page 5, Paragraph 3; Page 6, Paragraph 4; Page 7, Paragraph 1; Supplemental Table 2; changes highlighted in Blue).

 Major points:

  1. The primary outcome was time to first stroke recurrence requiring a hospital visit during follow-up. However, all-cause death can be competing risk. This should be considered in statistical analysis and needed to revise.
  2. For survival analyses, is day zero blood sampling? 

Response: We thank the reviewer again for his/her helpful suggestions.

  1. The competing risk analysis was performed accordingly (Page 5, Paragraph 3; Page 6, Paragraph 4; Page 7, Paragraph 1; Supplemental Table 2; changes highlighted in Blue).
  2. The length between the onset of stroke and blood sampling date had a mean of 6.3 years (standard deviation: 3.3). As a sensitivity analysis, we further adjusted for this variable in the survival analysis and found similar results (Page 5, Paragraph 3; Page 6, Paragraph 4; Page 7, Paragraph 1; Supplemental Table 2; changes highlighted in Blue).

Round 2

Reviewer 2 Report

Appropriate fixes have been made.